# Human Dental Pulp Cells form Spheroids in the Presence of Serum When Seeded on a Low-Attachment Cultural Surface

Linna Guo [1,2,*,†], Ziang Zou [1,3,†], Marcus Freytag [2], Reinhard E. Friedrich [2], Philip Hartjen [2], Martin Gosau [2], Ralf Smeets [2] and Lan Kluwe [2]

1   Department of Stomatology, The Second Xiangya Hospital, Central South University,
    Changsha 410011, China; zouziang_001@yeah.net
2   Department of Oral and Maxillofacial Surgery, University Hospital Hamburg-Eppendorf,
    20246 Hamburg, Germany; freytag@uke.de (M.F.); r.friedrich@uke.de (R.E.F.); p.hartjen@uke.de (P.H.);
    m.gosau@uke.de (M.G.); r.smeets@uke.de (R.S.); kluwe@uke.de (L.K.)
3   Department of Gynecology and Obstetrics, The Third Xiangya Hospital, Central South University,
    Changsha 410013, China
*   Correspondence: guolinna66@csu.edu.cn
†   These authors contributed equally to this work.

**Abstract:** Spheroid formation is a characteristic feature of stem/progenitor cells. Under a serum-free cultural condition, human dental pulp cells can form spheroids. In the present study, we report that these cells can also form spheroids in the presence of serum when seeded on a low-attachment cultural surface. Dental pulp cells derived from three teeth were seeded with surface densities $10^3$–$10^5$/cm$^2$ in wells of low attachment and standard cultural plates. Fibroblasts were also seeded onto a low-attachment surface as a comparison. The growth of single spheroids of pulp cells was observed for 7 days. Pulp cells in spheroids and cells attached to the low-attachment surface were separated and further expanded on standard cultural surface in the monolayer and studied for their viability and osteogenic differentiation comparatively. In all three cultures of primary human dental pulp cells on low attachment surface, spheroids formed one day after seeding and grew in size over the 7 days of observation. The optimal seeding density for spheroids was around $10^4$ cells/cm$^2$ ($10^5$ cells/mL). Expanded pulp cells from the spheroids exhibited lower viability but higher osteogenic differentiation potential compared to pulp cells expanded from those attached to the surface of the low attachment plate. Human dental pulp cells have the specific capacity to forms spheroids when seeded on a low-attachment surface and may enable selection of a subpopulation with stronger differentiation potential and may also provide a strategy for culturing these cells in a three-dimensional organization without scaffolds.

**Keywords:** dental pulp cells; spheroids; low attachment plate; stem cell; differentiation

## 1. Introduction

Human dental pulp cells exhibit features of mesenchymal stem cells, especially their multipotent differentiation into multiple lineages [1,2]. We have also successfully induced adipogenic and osteogenic differentiation from dental pulp cells [3] These cells can be relatively easily obtained from the roots of extracted teeth, providing an autologous resource for tissue engineering in regenerative medicine.

Spheroid formation is another characteristic feature of stem cells. Human dental pulp cells can form spheroids under a serum-free cultural condition [4]. Cells with stem-cell features are found in the core of these spheroids and increase in number with time [5]. Spheroids also provide a possibility of culturing cells in a three-dimensional (3D) organization without scaffolds [6]. Compared to monolayer cultures, 3D cultures are more natural and closer to the natural conditions in the human body [7].

Usually, hydrophilic surfaces are used for culturing cells which enhance cell attachment. In contrast, cells do not attach well to hydrophobic surfaces, which are also termed as "low attachment" surfaces. We have observed that when the dental pulp cells were seeded onto low-attachment surfaces, some cells did not attach to the surface and form spheroids, despite the fact that the medium contained 15% serum.

In the present study, we explored this observation. We examined whether or not the spheroid-forming capacity in the presence of serum is a specific feature of the dental pulp cells by including fibroblasts as a comparison. We then followed single spheroids of pulp cells for up to 7 days. Finally, pulp cells were expanded from spheroids and examined for their viability and osteogenic differentiation in comparison to cells expanded from pulp cells, which remained as a monolayer on a low-attachment surface.

## 2. Materials and Methods

### 2.1. Cell Culture

Human dental pulp cells were cultured from a total of 3 anonymized wisdom teeth of adults. These teeth were extracted for medical reasons at our Department of Oral and Maxillofacial Surgery, Eppendorf Medical Center Hamburg-Eppendorf, Germany, and would have been discarded as biowaste. With written consent from the patients, the teeth were saved for the study. The outgrowth method was used to culture dental-pulp cells [8]. Briefly, the teeth were broken with a hammer, and the pulp tissues were taken out and cut into small pieces which were placed onto the cultural surface in wells of 6-well plates. After approximately 1 week, cells started migrating from the explants and grew in the monolayer. The characterization of the pulp cells has been proven in our previous study [3].

### 2.2. Spheroidal Culture

Upon subconfluence, cells were trypsinized and resuspended in MEM with 15% fetal bovine serum and antibiotics at approximately $3 \times 10^3$–$3 \times 10^5$ cells/mL. Next, 100 μL of the suspension was seeded into 96-plate wells of low attachment (Sarstedt AG & Co., Nümbrecht, Germany), giving surface densities of $10^3$, $7 \times 10^3$, $10^4$, $7 \times 10^4$ and $10^5$ cells/cm$^2$, and volume density of $3.3 \times 10^3$, $2.3 \times 10^4$, $3.3 \times 10^4$, $2.3 \times 10^5$ and $3.3 \times 10^5$ cells/mL.

For comparison, mouse fibroblast L929 and human gingival fibroblasts were also seeded onto the low-attachment surface at the same surface and volume densities as for the pulp cells.

Upon spheroid formation, the medium containing the floating spheroids was collected, diluted and added to a 96-well plate. Wells containing a single spheroid were identified, and the spheroids in them were followed for up to 7 days. This way, the effect of spheroids fusing and moving of adhesive cells to the spheroids could be excluded.

### 2.3. Expanding Cells from Spheroids and from Monolayer

The medium containing the floating spheroids were collected in a tube. The spheroids were dissociated with 0.05% trypsin for 5 min. The spheroid cells were then cultured as a monolayer on a standard cultural surface for expansion. After removing the medium, the remaining adhesive cells on the low-attachment surface were harvested using 0.05% trypsin and reseeded onto the standard cultural surface for expansion.

After expansion, the cells from spheroids and from monolayer were measured for their viability using a MTS assay (Promega). Briefly, a one-fifth volume of MTS solution was added to each well and the plates were incubated for 1–4 h at 37 °C. The absorbance at 490 nm was measured.

### 2.4. Osteo-Differentiation

Pulp cells expanded from spheroids and from adhesive cells as described above were harvested and seeded in wells of a 24-well plate. At approximately 80% confluence, differentiation was initiated by changing the media to DMEM/Hams F12 supplemented

with 10% human serum and components for inducing osteo-differentiation (50 µM ascorbic acid 2-phosphate and 10 mM β-Glycerophosphate). The media were refreshed every other day and the differentiation was continued for three weeks.

Alkaline phosphatase activity was measured using a p-nitrophenyl phosphate assay at 405 nm [9]. The enzyme activity values were normalized against the viability of the cells.

### 2.5. Statistical Analysis

Statistical analysis was performed using SPSS 21 (IBM, Armonk, NY, USA). Normal distribution of viability and ALP values were assessed using the skewness–kurtosis method. Afterward, all values were compared by one-way analysis of variance (ANOVA) and *t*-test. For all results, statistical significance of differences was set at $p < 0.05$.

## 3. Results

### 3.1. Spheroids Formation

In all three cultures of dental pulp cells derived from three unrelated teeth on a low-attachment surface, spheroids were observed on the next day of seeding for those with seeding density higher than $7 \times 10^3$ cells/cm$^2$ and $2.3 \times 10^4$ cells/mL (Figure 1). No spheroids formed for the culture with a seeding density of $10^3$/cm$^2$. The size of the spheroids varied. The number of spheroids increased with increasing seeding density. However, fusing of spheroids also increased with increasing seeding density. For example, at $10^5$ cells/cm$^2$, large aggregates with irregular forms were visible.

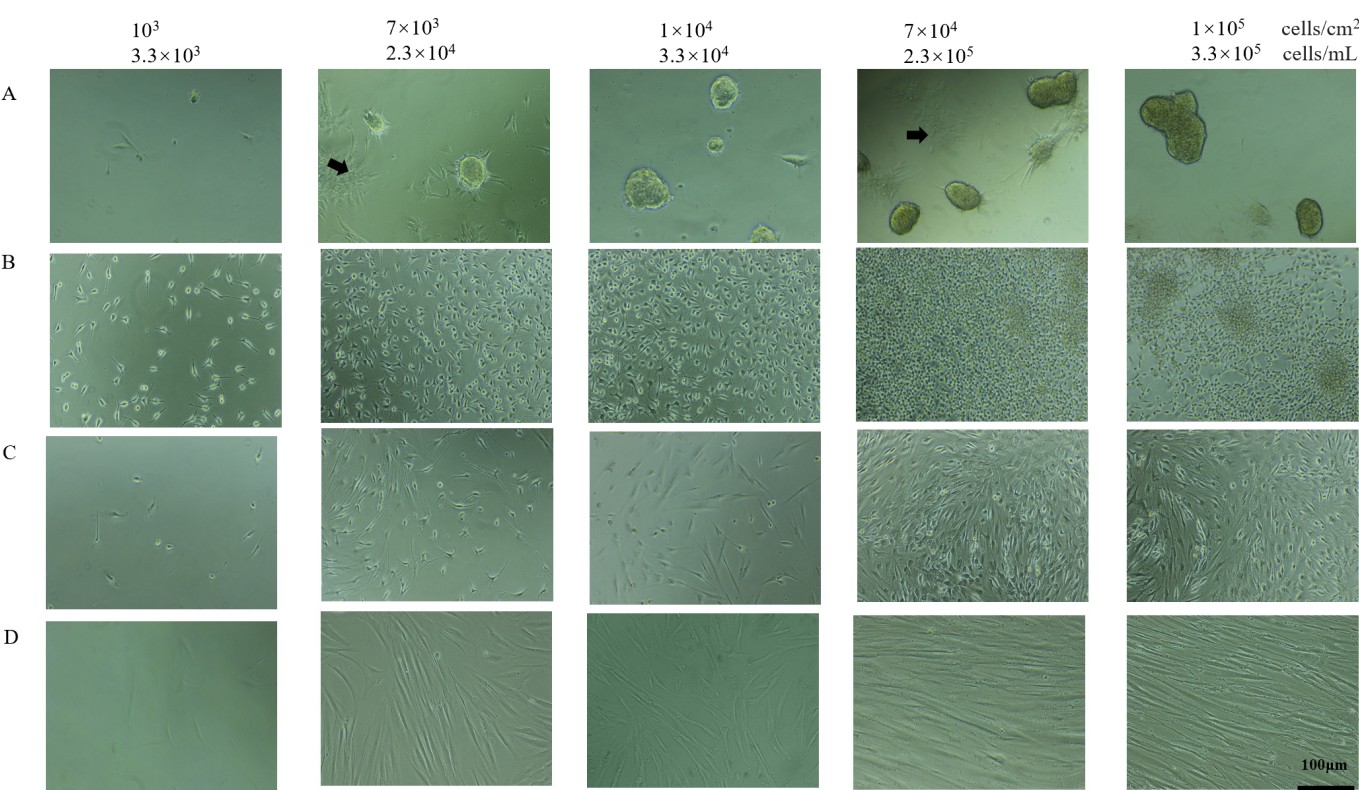

**Figure 1.** Spheroid formation from human dental pulp cells on the low attachment cultural surface one day after the seeding (**A**). No spheroids in mouse fibroblasts cells L929 (**B**) and human gingival fibroblasts (**C**) cultured on the same low-attachment surface. No spheroids in human dental pulp cells when cultured on standard culture surface (**D**). Cells were seeded at various surface (upper row, in cm$^2$) and volume (lower row, in mL) densities, as given on the top of the photos. Object magnification: 10-fold. Array: remaining adhesive cells in island-like structure.

The spheroids were not perfectly spheric but rather ellipsoid. However, the surface of the spheroids was smooth and shiny, and the bodies were translucent, differing from simple aggregates and indicating a certain kind of inter-cellular organization and structure. All the spheroids detached from the cultural surface and floated in the medium.

There were also pulp cells which remained adhesive to the low-attachment surface. However, these cells were not evenly distributed, but rather grew in island-like patches (Figure 1).

As a comparison, mouse fibroblasts cells L929 and human gingival fibroblasts were also cultured on the same low-attachment surface. However, no spheroids were observed in any of them, regardless of the seeding density. No islands of cells in monolayer were observed. The fibroblasts remained single, adhesive and were evenly distributed (Figure 1B,C). No spheroids were formed in pulp cells when seeded on a standard cultural surface (Figure 1D).

For better observation, some of the spheroids of the pulp cells on the low-attachment surface were transferred into new wells of a low-attachment surface in such a way that each well contained only one single spheroid. Some spheroids became obviously larger over the 7 days of observation (Figure 2). Since there were no adhesive cells and no other spheroids in the well, the increasing size of such spheroids likely reflects the growth of cells in the spheroids.

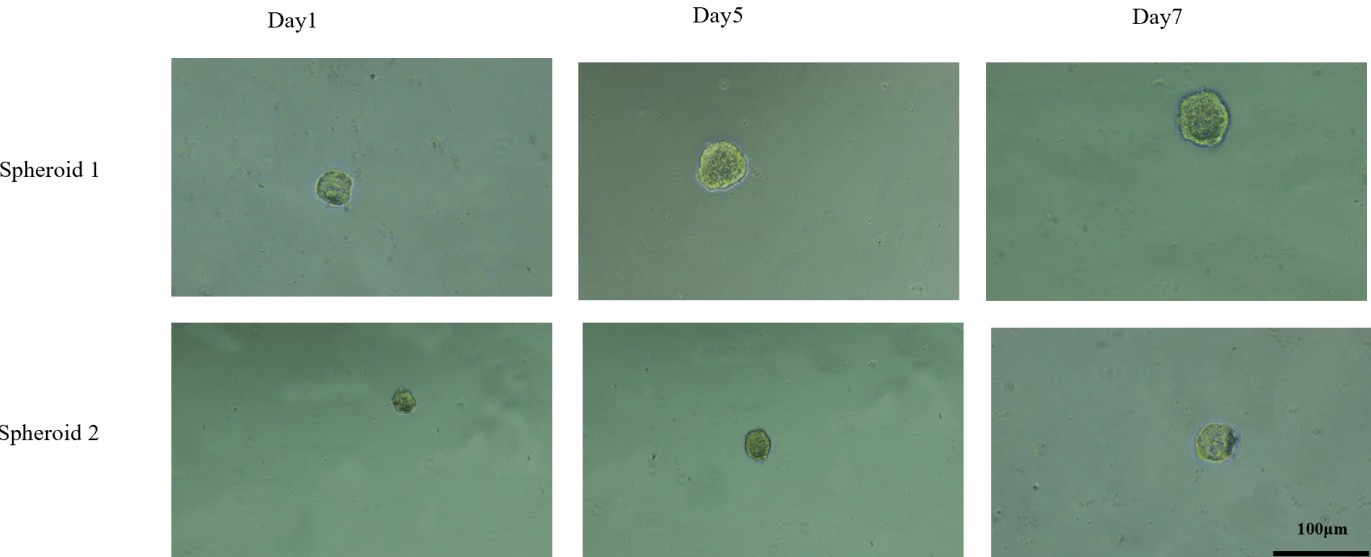

**Figure 2.** The growth of two single spheroids, each in one well, over 7 days. Some spheroids became obviously larger over the 7 days of observation. In the process of growth, the characteristics of the sphere can be kept unchanged, but the volume increases.

### 3.2. Compare Pulp Cells Expanded from Spheroids and from Monolayer

Pulp cells from spheroids exhibited significantly lower viability compared to pulp cells which were adhesive on the low-attachment surface after 7 days of seeding (Figure 3A). This was true for all three pulp cell cultures derived from three different teeth.

In contrast, after 21 days of osteogenic differentiation, significantly higher alkaline phosphatase activity ($p < 0.05$) was measured in the former, indicating higher osteogenic differentiation potential (Figure 3B). Again, this was true for all three pulp-cell cultures derived from different teeth.

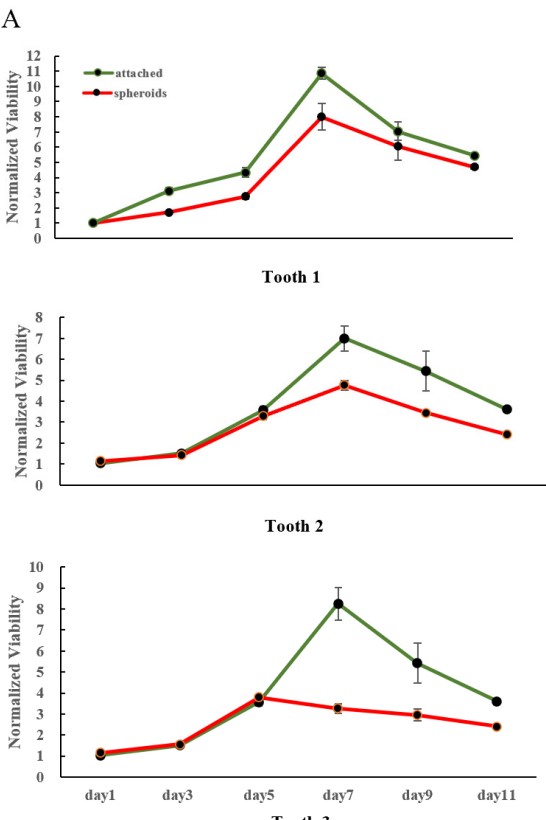

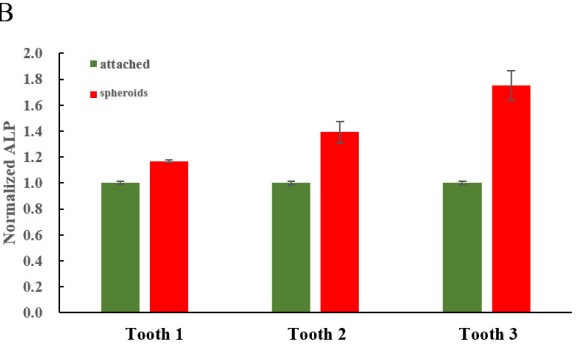

**Figure 3.** The viability (**A**) and alkaline phosphatase activity (**B**) of cells originated from spheroids and from cells remained adhesive on low-attachment surface. Pulp cells from spheroids exhibited significantly lower viability compared to pulp cells ($p < 0.05$). Alkaline phosphatase activities were measured 21 day later after the induction of osteogenic differentiation was initialized. The alkaline phosphatase activities were normalized by relative viability to day 1. After 21 days of osteogenic differentiation, significantly higher alkaline phosphatase activity was measured in the former. The same experiments were repeated for dental pulp cells from 3 unrelated teeth ($p < 0.05$).

## 4. Discussion

In the present study, spheroids of dental pulp cells were formed on low-attachment surfaces. In contrast, no spheroids were formed in fibroblasts cultured under the same condition. No spheroids were formed in pulp cells when cultured on a standard surface. Therefore, spheroid formation depends on both the cell type and the feature of the cultural surface. Pulp cells expanded from spheroids exhibited higher potential in osteogenic differentiation than the adhesive cells that were grown.

Low-attachment surfaces do not provide optimal conditions for cells to grow on it. Cells are then forced to move to each other and organize themselves in a kind of self-assembly. When such self-assembly has a spherical form, a translucent body and smooth and shining boundary, the cells are defined as spheroids [10,11]. However, not all cells can form spheroids when cultured using low-attachment surfaces. The ability to form spheroids is a specific feature of stem/progenitor cells. Stronger stemness corresponds to a higher quality of spheroids, while the embryonic sphere is the most perfect one. We observed spheroids only in the pulp cells but not in the fibroblasts, though they all grew on the low-attachment surface. This finding is in concordance with the specific stem-cell feature of the dental pulp cells.

In the present study, we seeded dental pulp cells from $10^3$ to $10^5/cm^2$. Generally, spheroids form a better higher-cell density. However, we found that too-high cell density will also lead to irregular form and fusion of the spheroids. At too-high density, the inter-cellular organization and structure may not be optimal.

Morintani et al. formed spheroids of human periodontal ligament mesenchymal stem cells with 10% FBS in microwell chips and observed the growth of these spheroids. However, after 72 h of observation, the size of the spheroid decreased over time [12]. In the present study, the diameter of the spheroid increased with time, indicating that the spheroid can grow under certain conditions. Whether the spheroidization method and culture conditions of the present study can better simulate the spheroid environment in vivo without scaffold still needs to be further explored. This study provokes some new thoughts for the exploration of spheroidization methods.

Pulp cells expanded from spheroids had lower viability and higher potential in osteogenic differentiation compared to cells which remained adhesive on the low-attachment surface, both of which were in concordance with stronger stemness of cells from spheroids. However, in the present study, spheroids from the low-attachment surface were dissociated and the cells were expanded in the monolayer on a standard cultural surface. They may lead to loss of some stemness. In future study, differentiation can be initiated directly from spheroids on the low-attachment surface without dissociating them.

## 5. Conclusions

Spheroids can be easily obtained from human dental pulp cells by seeding them onto low-attachment cultural surfaces. Cells in the spheroids likely have stronger stemness and exhibited higher potential in differentiation. In addition, spheroids provide a strategy for natural 3D self-assembly of the cells, which may have application in tissue engineering.

**Author Contributions:** L.G.: study conception and design, experimental operation, data collection, analysis and interpretation, critical editing of the manuscript; L.K. and P.H.: study conception and design, data analysis and interpretation, critical editing of the manuscript; Z.Z. and R.E.F.: study conception and design, experimental operation, data collection, and analysis; M.F.: experimental operation, data collection, and analysis; M.G. and R.S.: study conception, discussion, and critical editing. All authors have read and agreed to the published version of the manuscript.

**Funding:** L.G. and Z.Z. were supported by the China Scholarship Council (No. 201806370248; No. 201806370249).

**Institutional Review Board Statement:** Not applicable.

**Informed Consent Statement:** All patients signed written informed consent forms for using anonymized waste-specimen.

**Data Availability Statement:** The data that support the findings of this study are available within the manuscript. Additional information can be provided from the corresponding author upon reasonable request.

**Acknowledgments:** The present study would not have been possible without the participation of the patients and healthy volunteers.

**Conflicts of Interest:** The authors deny any conflict of interest.

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
