# Peer review of "Human Dental Pulp Cells form Spheroids in the Presence of Serum When Seeded on a Low-Attachment Cultural Surface"

_processes, doi:10.3390/pr10051021_

Round 1

Reviewer 1 Report

This study describes the formation of spheroid from human dental pulp cells culturing on low attachment surface. The authors also noted that the cells from spheroids showed lower viability and higher osteogenic differentiation compared to the cells attached to the low attachment surface. This is unique and interesting study design, however, there are several concerns with study that need addressing, as indicated below.

  1. In the abstract, the authors described that the optimal density of spheroids was about 104 cells/cm2(105cells/ml). This point need to be explained more detailed in the discussion.
  2. The authors noted that cells from spheroids have stronger stemness. Are these cells stem cells ? The FACS analysis of the expression profile needs to be shown.
  3. Low attachment surface can form spheroids with human dental pulp cells and can’t form those with L929 and human gingival fibroblasts. What is the reason for this difference?
  4. Page 4 line 136-137: Significant differences (p<0.05) are shown in the Alkaline phosphatase activity assay. However, statistical analysis methods and groups with significant differences were not described in the materials and methods, figure 3 and figure 3 legends. Please check these.
  5. Figure1 legend (page 3 line 117): Although it is described as Array, is it an arrow? Please check it.
  6. Figure2 legend (page 4 line 131): The description is difficult to understand in detail. Please be a little more specific.
  7. Figure3: The description of the X-axis is lacked in Teeth 1 graph and Teeth 3 graph on Figure3A. Also, the description and order are different between Figure3A and Figure3B. Teeth or Tooth? Please check these.

Author Response

Response to Reviewer 1:

  1. In the abstract, the authors described that the optimal density of spheroids was about 104 cells/cm2(105cells/ml). This point need to be explained more detailed in the discussion.

Response: Thank you for your suggestion. We explained more detailed concerning the optimal density of spheroids in the discussion section (line178-181, page 6).

  1. The authors noted that cells from spheroids have stronger stemness. Are these cells stem cells? The FACS analysis of the expression profile needs to be shown.

Response: Thank you for your suggestion. We did not carried out FACS analysis of the spheroids. Our osteogenic differentiation data suggest that cells from the spheroids may have stronger stemness. But this is a very good suggestion. We can conduct FACS analysis in future research for further exploration concerning the stemness of cells from spheroids.

  1. Low attachment surface can form spheroids with human dental pulp cells and can’t form those with L929 and human gingival fibroblasts. What is the reason for this difference?

Response: Thank you for your suggestion. Spheroid formation is a special feature of stem cells or stem cell-like cells. However, under certain very special condition, also non-stem cells will be forced to form some sort of spheroid-like aggregates. We included fibroblast as a comparison to show that not all cells form spheroids under this given condition, but rather that spheroid formation is a special feature dental pulp cells reflecting their stem cell like features.

  1. Page 4 line 136-137: Significant differences (p<0.05) are shown in the Alkaline phosphatase activity assay. However, statistical analysis methods and groups with significant differences were not described in the materials and methods, figure 3 and figure 3 legends. Please check these.

Response: Thank you for your suggestion. The description of statistical analysis was added in the materials and method section (line98-101, page 3) and also in figure 3 legends (line154-155, line 157-160, page 6)

  1. Figure1 legend (page 3 line 117): Although it is described as Array, is it an arrow? Please check it.

Response: Thank you for your suggestion. we added the array in figure 1 A (7x103 and 7x104/cm2).

  1. Figure2 legend (page 4 line 131): The description is difficult to understand in detail. Please be a little more specific.

Response: Thank you for your suggestion. We added some specific description of figure 2 (line 140-142, page 4)

  1. Figure3: The description of the X-axis is lacked in Teeth 1 graph and Teeth 3 graph on Figure3A. Also, the description and order are different between Figure3A and Figure3B. Teeth or Tooth? Please check these.

Response: Thank you for your suggestion. I'm sorry for our negligence. We have revised accordingly (Figure3, line 152, page 5).

Reviewer 2 Report

An extensive spell-check is needed, including the text in the figures. The number of references is rather low, it would be appropriate to extend it, if possible. It would also be helpful to add more details on how the results may be used further.

Author Response

Response to Reviewer 2:

An extensive spell-check is needed, including the text in the figures. The number of references is rather low, it would be appropriate to extend it, if possible. It would also be helpful to add more details on how the results may be used further.

Response: Thank you for your suggestion. I appreciate your carefully reading and valuable suggestions. we carefully check the whole manuscript and correct some errors. We also added some relative references accordingly. We also added more details on how the results may be used further in the discussion section (line190-192, line 196-197, page 6).

Round 2

Reviewer 1 Report

The manuscript is now much improved.

Please check the figure 3 (page 5) because two figures are shown in revision.

Author Response

Dear reviewer:

Thank you for you suggestion. I deleted the wrong figure in page 5.